Comparative analysis of combined phosphorus and drought stress-responses in two winter wheat

Zhang Xiangchi
Li Chao
Lu Weidan
Wang Xiaoli
Ma Bin
Fu Kaiyong
Li Chunyan lichunyan@shzu.edu.cn
Li Cheng lichean_79@aliyun.com
Shihezi University , Shihezi , China
Fukushima Atsushi
Electronic publication date: 2022 Sep 22
Publication date: 2022
Volume: 10
Electronic Location ID: e13887
Received 2021 Nov 29; Accepted 2022 Jul 21
Copyright: © 2022 Zhang et al.
Copyright year: 2022
Copyright holder: Zhang et al.
License: This is an open access article distributed under the terms of the Creative Commons Attribution License, which permits unrestricted use, distribution, reproduction and adaptation in any medium and for any purpose provided that it is properly attributed. For attribution, the original author(s), title, publication source (PeerJ) and either DOI or URL of the article must be cited.
License URL: https://creativecommons.org/licenses/by/4.0/

Keywords: Drought stress, Phosphorus, Root, Wheat, Abiotic stress

Funding: National Natural Science Foundation of China 31860335, 31860337, 31360334, 31560389 Young Innovator Cultivating Project of Shihezi University CXRC201703 Specific Project for Breeding of Shihezi University YZZX201902 This study was financially supported by the National Natural Science Foundation of China (31860335, 31860337, 31360334, and 31560389), the Young Innovator Cultivating Project of Shihezi University (CXRC201703), the Specific Project for Breeding of Shihezi University (YZZX201902). The funders had no role in study design, data collection and analysis, decision to publish, or preparation of the manuscript.

==============================
Phosphorus stress and drought stress are common abiotic stresses for wheat. In this study, two winter wheat varieties “Xindong20” and “Xindong23” were cultured in a hydroponic system using Hoagland nutrient solution and treated with drought stress under conventional (CP: 1.0 mmol/L) and low (LP: 0.05 mmol/L) phosphorus levels. Under drought stress, the root growth was better under LP than under CP. Under LP, root phosphorus content was increased by 94.2% in Xindong20 and decreased by 48.9% in Xindong23 at 3 d after re-watering, compared with those at 0 d under drought stress. However, the potassium (K) content was the highest among the four elements studied and the phosphorus (P) and calcium (Ca) content were reduced in the root of the two varieties. Under CP, the zinc (Zn) content was higher than that under LP in Xindong23. The GeneChip analysis showed that a total of 4,577 and 202 differentially expressed genes (DEGs) were detected from the roots of Xindong20 and Xindong23, respectively. Among them, 89.9% of DEGs were involved in organelles and vesicles in Xindong20, and 69.8% were involved in root anatomical structure, respiratory chain, electron transport chain, ion transport, and enzyme activity in Xindong23. Overall, LP was superior to CP in mitigating drought stress on wheat, and the regulatory genes were also different in the two varieties. Xindong20 had higher drought tolerance for more up-regulated genes involved in the responses compared to Xindong23.

Introduction

Wheat is one of the most important food crops in the world, and its growth is affected by many environmental factors. Phosphorus (P) is one of the essential nutrients for wheat growth. However, the distribution of soil P is uneven worldwide. Moreover, drought stress is very common in arid and semi-arid regions. Therefore, in some arid and semi-arid regions, P starvation and drought stress simultaneously threaten the growth of crops.

Under the P starvation condition, plants enhance P utilization by increasing P uptake and P utilization efficiency (Vance, Uhde-Stone & Allan, 2003). The changes in root structure, bio-availability of soil inorganic P, high-affinity P transporter, and some metabolic pathways have also been detected in plants to enhance P utilization (Liu et al., 2010). Besides, previous studies found that the mycorrhizal symbiosis is closely associated with P transporters and P utilization, and 80% of terrestrial plant increase nutrient uptake through fungal hyphae (Harrison, Dewbre & Liu, 2002; Paszkowski et al., 2002). Katrin et al. (2019) showed that P utilization could be enhanced by secreting organic acids, protons, acid phosphatases, and ribonuclease by plant roots.

Drought stress could suppress the growth of plant roots, while plants may change their morphology to adapt to the drought stress (Deak & Malamy, 2005), such long and dense roots in drought-tolerant plant (Hurd, 1975). Drought stress affects plant photosynthesis, membrane lipids, active oxygen metabolism, carbon and nitrogen metabolism, and osmotic adjustment. Previous studies showed that large quantities of reactive oxygen species were accumulated in plant roots under drought conditions, causing membrane lipid peroxidation (Moller, Jensen & Hansson, 2007; Cruz de Carvalho, 2008; Anjum et al., 2011). Therefore, maintaining high antioxidant enzyme activity is essential for enhancing plant drought tolerance (Sharma & Dubey, 2005; Türkan et al., 2005). It has been reported that the structure of the root and canopy of plants are affected by the distribution of dry matter (Hermans et al., 2006). Under drought conditions, plants produce more root branches by sucrose metabolism to enhance drought tolerance (Macgregor et al., 2008). In addition, silicon has also been proven to be one of the important elements involving in the stress tolerance of plants (Epstein, 1999; Savant et al., 1999). Silicon could affect the membrane permeability of rice under drought stress (Agarie et al., 1998). Adams, Kendall & Kartha (1990) reported that trehalose could act as a penetration protectant under drought conditions.

Water deficit affects the uptake of nutrients by crops, and the deficiency of nutrients may lead to the decrease of water use efficiency, thus affecting crop yield and quality (Manske et al., 2001; Condon et al., 2004; Wang et al., 2009). Wu et al. (2018) found that increasing nitrogen and P application could minimize the effects of drought stress on bamboo growth through increasing photosynthetic rate and membrane integrity. Root hair plays a crucial role in relieving water and P stress. Nussaume et al. (2011) found that P could be obtained by plant roots through specific transporters, such as high-affinity phosphate transporters. Besides, Kellermeier et al. (2014) showed that the nutrients in plants had a significant influence on the expression of genes that regulate root structure. For instance, Jain et al. (2013) showed that P starvation response signaling pathway and the expression of some target genes, especially Pht1;1 in the root, appear to be suppressed under zinc-deficient conditions. In ectomycorrhizal symbiosis, P and K+ jointly affect membrane transport and transferred to host plants (Garcia et al., 2014), while high K+ affects the uptake of P by Arabidopsis (Ródenas et al., 2019). Matthus et al. (2019) found that when P starvation occurred, Arabidopsis root tips showed a strong spatiotemporal [Ca2+] cyt response to eATP. Therefore, the change of Ca2+ content in plants could indicate the growth status of plant roots and the response to stress (Behera et al., 2018; Manishankar et al., 2018). However, the molecular mechanisms of plants responses to combined drought and P stress have not been reported.

High-throughput technologies, such as large-scale parallel sequencing and gene chips, have been applied in molecular biological researches on the response of model plants to P starvation. At present, many genes involved in the responses to P starvation have been identified. They play an essential role in regulating P homeostasis plants (Zhou et al., 2015). In addition, the transcriptional characteristics of wheat responses to temperature, drought, and salt stress could also be detected by microarray technology. However, at present, there are few studies on the transcriptomic responses of wheat combined phosphorus and drought stress. Therefore, in this study, the effects of different P application rates (LP and CP) on the root growth of two wheat varieties (Xinong20 and Xinong23) under drought stress were determined through the detection of root physiological information and GeneChip. This study aimed to explore the physicochemicaland transcriptomes responses of wheat roots to two P levels under drought stress. We speculated that the gene expression in wheat was changed accordingly with the P content and drought stress.

Materials and Methods

Plant material and cultivation

The winter wheat varieties “Xindong20” and “Xindong23” provided by the Winter Wheat Research Group of the Agricultural College of Shihezi University were used in this study. “Xindong20” has higher drought tolerance and P utilization efficiency than the “Xindong23”. To obtain complete and undamaged root systems, wheat plants were cultured in hydroponic system using Hoagland nutrient solution, in growth cabinets (JNYQ, RXZ-300B-LED, China) with an 18 h day-length under photosynthetically active light (250 µmole m−2 s−1). The temperature during the day was 25 °C and at night it was 20 °C. Plump wheat seeds were sterilized in 1‰ HgCl2 solution for 30 s, rinsed with distilled water, and placed evenly in a petri dish lined with wet filter paper. The petri dishes with seeds were placed in an incubator in the dark to promote germination. Distilled water was regularly replenished to keep moist. After 1 week, the seedlings with similar growth were selected and transferred to a black plastic box to maintain roots in darkness. Each box contained the same volume of Hoagland nutrient solution. The boxes were divided into conventional P level (CP: 1.0 mmol/L) and low P level (LP: 0.05 mmol/L). KCl was used to replace part of KH2PO4 under LP. The nutrient solution was changed every 3 days. After 14 d of culture, 15% (w/v) PEG-6000 (osmotic potential: −0.32 MPa) was added into the Hoagland nutrient solution (both CP and LP) to simulate drought stress on wheat seedlings for 7 d. Then the wheat seedlings were cultured with Hoagland nutrient solution without PEG-6000 under CP and LP for 3 d. The roots tissues were sampled at 0 d, 3 d, 5 d, 7 d under drought stress (DS 0 d, 3 d, 5 d, and 7 d) and 3 d after re-watering (RW 3 d). Part of the samples at DS 7 d were quickly frozen in liquid nitrogen and stored at −80 °C.

Root morphological scanning

Fresh roots were soaked in tap water for 10 min, and then rinsed with running water. The roots were completely spread out to avoid overlapping. The plant image scanner (Wanshen LS-A, Phantom 9850XL PLUS, China) was used for scanning, and root images were stored in a computer. The root analysis system (WinRHIZO 2009 system) was used to determine the total root length, total root surfarea, and the number of forks crossings, and root tips.

Root absorption area measurement

Surface water on fresh roots were absorbed by the filter paper. The root absorption area was determined according to the method described by Zhang (1990). Briefly, methylene blue solution (0.2 μmol/L) was transferred into three numbered small beakers. The volume of the solution in each beaker was about 10 times the volume of the root. The roots sampled at different dates were immersed in different beakers containing methylene blue solution for 1.5 min. After that 1 mL of the solution was taken from each of the three beakers and diluted 10 times. Distilled water was used for the control group. Absorbance was measured at 660 nm using a spectrophotometer (Shanghai Precision Scientific Instrument Co., Ltd., Shanghai, China).

Determination of root vitality

The root vitality was determined using a modified method (Kang & Saltveit, 2002). Wheat root samples (0.2 g) were transferred into a 50 mL centrifuge tube, and then 5 mL of TTC solution (0.4%, w/v) and 5 mL of salt buffer (1 mol/L, pH 7.0) were added. Before incubation for 1 h at 37 °C, the roots samples were completely immersed in the reaction solution. When the root samples turned red, the methanol-soaked method was used to completely whiten the apical segments (approximately 4–6 h). Absorbance was measured at 485 nm using a spectrophotometer (Shanghai Precision Scientific Instrument Co., Ltd., Shanghai, China).

Measurement of water content

The water content of wheat seedlings was determined by the method described by Gao (2006). Briefly, seedlings were washed under running tap water and dried using filter paper. Leaves and roots were separated, weighed, dried at 105 °C for 30 min and then tried and at 70 °C for 4–5 h to constant weight. Three replicates were adopted and ten wheat seedlings were included in each replicate.

Measurement of total phosphorus content

The total phosphorus content was determined according to the method described by Bao (2000). Briefly, dried, and ground sample (0.2 g) was transferred in a 100 mL tube and digested by H2SO4–H2O2, until the solution was completely clear. Then, dinitrophenol indicator and 6 M NaOH were added to neutralize the solution, and 10 mL of ammonium vanadylmolybdate reagent were added finally. Absorbance was measured at 440 nm using a spectrophotometer (Shanghai Precision Scientific Instrument Co., Ltd.722G, Shanghai, China).

Total RNA extraction and microarray analysis of root tissue

Total RNA was extracted using a Fruit-mate (9192; Takara, Shiga, Japan) and RNAiso plus (9108; Takara, Shiga, Japan) kit, according to the manufacturer’s instructions. Then, the total RNA was quantified using NanoDrop ND-2000 (Thermo Fisher Scientific, Waltham, MA, USA), and the integrity of the RNA was detected by gel electrophoresis. The qualified samples were sent to Beijing Compass Biotechnology, China for gene chip analysis (three replicates). Sample labeling, chip hybridization, and washing were performed with reference to the standard protocol of the chip technology (3′ IVT Expression Arrays; Affymetrix, Santa Clara, CA, USA). First, the total RNA was reverse transcribed to generate a double-stranded cDNA, and the double-stranded cDNA was transcribed to generate biotin-labeled cRNA. The labeled cRNA was then fragmented and chip-hybridized. After washing and staining, the original images were obtained using an Affymetrix Scanner 3000 (7G; Affymetrix, Santa Clara, CA, USA).

Command Console software (version 4.0, Affymetrix, Santa Clara, CA, USA) was used to analyzeimages to get raw data, and then Expression Console software (version 1.4.1, Affymetrix, Santa Clara, CA, USA) was used for RMA normalization. The rationality of the experimental design and the uniformity of replicates were evaluated using principal component analysis (PCA). Then, a moderated t-test was performed to identify genes with differential expression. The p value 0.05 and fold change (|log2FC|) greater than one were used as the thresholds to determine the differentially expressed genes (DEGs). Additionaly, the AgriGO database (http://bioinfo.cau.edu.cn/agriGO/index.php) was used for the GO enrichment analysis of DEGs, and the Bonferroni correction method was used to adjust the p value to control the false discovery rate (FDR). GO items with FDR lower than 0.05 were considered significant and used to analyze the biological functions of DEGs.

The ionome analysis

Twenty seedlings with similar growth were randomly collected from each of the CP and LP treatments. The content of calcium (Ca), potassium (K), zinc (Zn), and phosphorus (P) in root samples were determined using Agilent ICP-OES 710 (GenTech, South San Francisco, CA, USA). The element standard solution (1,000 μg/mL) was measured and diluted with 2% nitric acid to prepare the standard curve. Briefly, the oven-dried sample of 0.1 g was put in a polytetrafluoroethylene beaker, and then 10 mL of nitric acid was added. The mixture was heated on a hot plate for 12 h, and then digested at 150 °C until 2–3 mL liquid remained. After cooling, the liquid was transferred to 20 mL volumetric flask, dilute with 2% nitric acid to constant volume. Meanwhile, the blank test was done.

qRT-PCR analysis

The rest of the wheat root samples were used for qRT-RCR reaction to verify the results of gene chip. Total RNA was extracted using the EASYspin Plus Complex Plant RNA Kit (RN53; AidLab, Beijing, China). The first-strand cDNA was synthesized using the EasyScript® One-Step gDNA Removal and cDNA Synthesis SuperMix (AE311-03; TransGen Biotech, Beijing, China). The wheat actin gene was selected as the endogenous control. According to the manufacturer’s instructions, the qRT-PCR reaction system was prepared with an SYBR Green I Master (LightCycler® 480 SYBR Green I Master; Roche, Indianapolis, IN, USA). A total of three replicates were performed in this experiment (each repetition contained 20 individuals). The primers used in this experiment were designed by Primer Premier 5 and DNAMAN software (Table 1). Each PCR reaction was repeated at least three times. Relative quantitative methods of ∆∆CT were used to analyze the changes in the number of selected genes.

Table 1 Primers used in qRT-PCR experiments.

Probe set ID	Forward primer (5′-3′)	Reverse primer (5′-3′)	
Ta.5385.1.S1_at	TCGACAACGCCTACTACACCA	TACGTCCATCACGAGTTCACC	
Ta.23044.1.A1_s_at	CGAACTCCAGGGCCACCTTC	GTCGACACCACCTCCGACAC	
Ta.2758.1.S1_at	TCGTCATGTTCGGCACCATCC	CGCTCACCCTGGACGTTCTC	
Ta.11120.1.S1_x_at	AACATCAACAGCACCAAGCC	TCAACAAAGCCTGCGAACGTC	
Ta.5456.1.A1_at	ACCAGAGGAAGGGATTCAGTG	GTGCGAATACAATACGATGCTG	

Data analysis

The data were analyzed using Excel software and Origin software (version 9.60, OriginLab, Northampton, MA, USA). Each measurement was repeated at least three times. Significance was determined by paired t-test at p < 0.05. Statistical charts were drawn by using Excel software (version 16.0, Microsoft, Redmond, WA, USA) and Adobe Photoshop software CC 2019 (version 20.0, Adobe, San Jose, CA, USA).

Results

Effects of two phosphorus levels on morphological and physiological characteristics of wheat root under drought stress

There were considerable morphological differences between Xindong20 and Xindong23 during drought stress both under LP and CP (Fig. 1). Because the root sampling was destructive, the variation was discontinuous. Only the differencees at the same date were considered. The root morphology changed with time in both varieties. However, compared with Xindong23, Xindong20 was more sensitive to the combined drought and phosphorus treatment. At DS 3 d, the total root length, total root surfarea, and the number of forks and crossings in Xindong20 were significantly greater under CP than under LP. However, at DS 5 d and DS 7 d, these were significantly lower under CP than under LP (Figs. 1A–1D). For Xindong23, the number of root crossings under CP was significantly different from that under LP on all sampling days except for DS 0 d (Fig. 1J). The root tips of Xindong20 was significantly more under LP than under CP at DS 5 d and DS 7 d (Fig. 1E). However, for Xindong23, the number of root tips was significantly lower under CP than under LP at DS 5 d and RW 3 d (Fig. 1K). The root absorption area of Xindong 20 was significantly greater under LP than under CP at DS 5 d, but that was significantly greater under CP than under LP at DS 7 d and RW 3 d (Fig. 1F). There was no significant difference in the root absorption area of Xindong20 between CP and LP on all sampling days (Fig. 1L).

Figure 1 Effects of two phosphorus levels on root morphological characteristics of wheat plants at seedling stage under drought stress.

CP: 1.0 mmol/L; LP: 0.05 mmol/L; Values are means ± standard deviation of three repetitions; One asterisk (*) and two asterisks (**) are significantly different at p < 0.05 and p < 0.01, respectively. XD20CP: Xindong20 under conventional phosphorus level; XD20LP: Xindong20 under low phosphorus level; XD23CP: Xindong23 under conventional phosphorus level; XD23LP: Xindong23 under low phosphorus level. (A and G) Effects of low phosphorus and conventional phosphorus levels on the total root length of two wheat varieties under drought stress. (B and H) Effects of low phosphorus and conventional phosphorus levels on the number of root forks of two wheat varieties under drought stress. (C and I) Effects of low phosphorus and conventional phosphorus levels on the root total surfarea of two wheat varieties under drought stress. (D and J) Effects of low phosphorus and conventional phosphorus levels on the number of root crossings of two wheat varieties under drought stress. (E and K) Effects of low phosphorus and conventional phosphorus levels on the number of root tips of two wheat varieties under drought stress. (F and L) Effects of low phosphorus and conventional phosphorus levels on the root absorption area of two wheat varieties under drought stress.

The changes of root vitality in Xindong20 and Xindong23 were similar across time under CP (Figs. 2A, 2F). Under drought stress, the root vitality of both varieties were significantly higher under CP than under LP on all sampling days except for DS 7 d.

Figure 2 Effects of two phosphorus levels on physiological characteristics of wheat plants at seedling stage under drought stress.

CP: 1.0 mmol/L; LP: 0.05 mmol/L; Values are means ± standard deviation of three repetitions; one asterisk (*) and two asterisks (**) are significantly different at p < 0.05 and p < 0.01, respectively. XD20CP: Xindong20 under conventional phosphorus level; XD20LP: Xindong20 under low phosphorus level; XD23CP: Xindong23 under conventional phosphorus level; XD23LP: Xindong23 under low phosphorus level. (A and F) Effects of low phosphorus and conventional phosphorus levels on the root vitalety of two wheat varieties under drought stress. (B and G) Effects of low phosphorus and conventional phosphorus levels on the water content in root of two wheat varieties under drought stress. (C and H) Effects of low phosphorus and conventional phosphorus levels on the water content in shoot of two wheat varieties under drought stress. (D and I) Effects of low phosphorus and conventional phosphorus levels on the total phosphorus content in root crossings of two wheat varieties under drought stress. (E and J) Effects of low phosphorus and conventional phosphorus levels on the total phosphorus content in shoot of two wheat varieties under drought stress.

The root water content of both varieties were significantly higher under CP than under LP on all sampling days. The root water content of Xindong20 was gradually decreased from DS 3 d to DS 5 d, and then it was gradually increased (Fig. 2B). However, the root water content of Xindong23 was lowest at DS 7 d, and then gradually increased (Fig. 2G). It should be noted that under LP, the change of root water content of Xindong23 was more significant than that of Xindong20 from DS 7 d to RW 3 d. The shoot water content of Xindong20 was higher under LP than under CP from DS 3 d to RW 3 d. The changes of shoot water content in in two wheat varieties under CP were almost consistent with those under LP (Figs. 2C, 2H).

The changes of total P content in root and shoot of Xindong20 was different from those of Xindong23 (Figs. 2D, 2E, 2I and 2J). For Xindong20, from DS 0 d to RW 3 d, the total P content of root and shoot were significantly higher under CP than under LP (Figs. 2D, 2E). However, those of Xindong23 were significantly higher under CP than under LP on all sampling days except for DS 5 d (Figs. 2I, 2J).

Identification of differential gene expression

The cluster analysis and PCA of DEGs that the experimental samples were good (Fig. 3). In this study, 4,577 and 202 DEGs (p < 0.05 & |logFC| > 1) were detected from the roots of Xindong20 and Xindong23, respectively. Among them, 3,207 up-regulated genes and 1,370 down-regulated genes were found in Xindong20, accounting for 70.07% and 29.93% of DEGs, respectively; 55 up-regulated genes and 147 down-regulated genes were found in Xindong23, accounting for 27.23% and 72.77% of the total DEGs, respectively. The number of DEGs in Xindong20 root was 22.7 times that of Xindong23, and the number of up-regulated genes of Xindong20 was 1.3 times that of the down-regulated gene (Table 2). According to Gene Ontology (GO) enrichment analysis, the DEGs of Xindong20 were mainly divided into two categories including biological processes and cell components (Fig. 4). However, the DEGs of Xindong23 were mainly divided into three major categories including biological processes, cellular components, and molecular functions. There was a common category (regulation process-mycelium development) for the two varieties. In this category, 29 genes associated with mycelium development were all down-regulated in Xindong20. In Xindong23, among the 14 genes associated with mycelial development, were seven up-regulated and seven down-regulated (Tables 3 and 4). In addition, 13 common DEGs were found in the two varieties.

Figure 3 Hierarchical cluster and principal component analysis of differentially expressed genes in the root of Xindong20 and Xindong23 at 7 d drought stress under CP and LP treatments.

(A) Hierarchical cluster analysis of differentially expressed genes in the root of Xindong20 at 7 d drought stress under CP and LP treatments. (B) Hierarchical cluster analysis of differentially expressed genes in the root of Xindong23 at 7 d drought stress under CP and LP treatments. (C) Principal component analysis of differentially expressed genes in the root of Xindong20 and Xindong23 at 7 d drought stress under CP and LP treatments. XD20CP: Xindong20 under conventional phosphorus level; XD20LP: Xindong20 under low phosphorus level; XD23CP: Xindong23 under conventional phosphorus level; XD23LP: Xindong23 under low phosphorus level.

Table 2 Numbers of differentially expressed genes (DEGs) in two wheat genotypes.

The number of differentially expressed genes were from the comparison between LP and CP treatment in the wheat root subjected to drought stress for 7 d, respectively.

Genotypes	Gene category	No. of genes	Percentage of total DEGS (%)	
Xindong20	Up	3,207	70.07	
	Down	1,370	29.93	
Xindong23	Up	55	27.23	
	Down	147	72.77	

Figure 4 Gene Ontology analysis of differentially expressed genes in the root of Xindong20 and Xindong23 at 7 d drought stress under CP and LP treatments.

(A) Gene Ontology analysis of differentially expressed genes in the root of Xindong20; (B) Gene Ontology analysis of differentially expressed genes in the root of Xindong23.

Table 3 The number of up-regulated and down-regulated probes in different GO terms in the root of Xindong20.

Term_type	Go-term	Up-regulated probe	Down-regulated probe	
B	DNA packaging	25	3	
B	Nucleosome assembly	22	3	
B	Nucleosome organization	22	3	
B	Chromatin assembly	22	3	
B	Protein-DNA complex assembly	22	3	
B	Cellular nitrogen compound metabolic process	51	29	
B	Chromatin assembly or disassembly	24	3	
B	Chromosome organization	29	5	
B	Mycelium development	0	29	
B	Chromatin organization	25	4	
C	Membrane-bounded vesicle	218	114	
C	Cytoplasmic membrane-bounded vesicle	218	114	
C	Vesicle	218	114	
C	Cytoplasmic vesicle	218	114	
C	Nucleosome	19	3	
C	Protein-DNA complex	19	3	
C	Chromosomal part	25	6	
C	Chromatin	24	3	
C	Chromosome	27	6	
C	Intracellular organelle	666	284	
C	Organelle	666	284	
C	Intracellular membrane-bounded organelle	647	240	
Note:

B, Biological Process; C, Cellular Component.

Table 4 The number of up-regulated and down-regulated probes in different GO terms in the root of Xindong23.

Term_type	Go-term	Up-regulated probe	Down-regulated probe	
B	Mycelium development	7	7	
B	Mitochondrial ATP synthesis coupled electron transport	0	9	
B	ATP synthesis coupled electron transport	0	9	
B	Respiratory electron transport chain	0	9	
B	Oxidative phosphorylation	3	11	
B	Alditol metabolic process	7	0	
B	Glycerol metabolic process	7	0	
B	Mitochondrial electron transport, cytochrome c to oxygen	0	5	
B	Ion transport	5	16	
B	Electron transport chain	0	9	
B	Oxidation reduction	1	9	
B	Anatomical structure development	11	14	
M	Transferase activity, transferring alkyl or aryl groups	7	6	
M	Nicotianamine synthase activity	0	5	
M	Glycerophosphodiester phosphodiesterase activity	4	0	
C	Mitochondrial respiratory chain	1	8	
C	Respiratory chain complex IV	0	5	
C	Respiratory chain	1	9	
C	Mitochondrial respiratory chain complex I	0	4	
C	Mitochondrial respiratory chain complex IV	0	4	
C	Mitochondrial membrane part	1	8	
Note:

B, Biological Process; C, Cellular Component; M, Molecular Function.

Functional categories of differentially expressed genes

For the DEGs in Xindong20, in the biological process category, about 40% of these genes were associated with the assembly of complex (including DNA packaging, nucleosome assembly, chromatin assembly, protein-DNA complex assembly, and chromatin assembly or disassembly), about 36% were associated with the organization of complex (including mycelium development, nucleosome organization, chromosome organization, and chromatin organization), and about 24% were associated with cellular nitrogen complex metabolic processes (Fig. 5A). The up-regulated genes accounted for 81% of the total in the biological process, but the genes associated with the mycelial development were down-regulated in Xindong20 (Table 3).

Figure 5 Effects of two phosphorus levels on functional category distribution of differentially expressed genes in Xindong20 and Xindong23 under drought stress.

(A and B) Biological process and cellular component in Xindong20; (C–E) biological process, cellular component and molecular function in Xindong23.

In the cellular component category, the DEGs mainly regulated the organelles and vesicles in Xindong20. Among them, about 65% and 32% were associated with organelle (70% was up-regulated) and vesicle (66% was up-regulated), respectively. Moreover, about 3% were associated with nucleosome, chromatin, chromosome, and protein-DNA complexes (Fig. 5B).

The number of DEGs in Xindong23 root was less than that in Xindong20 root under LP and CP at DS 7 d. In the biological process category, 30% of DEGs were associated with electron transport, 18% were associated with the development of root anatomical structure, 15% were associated with the ion transport, and 10% were associated with the alcohols material metabolism (Fig. 5C). Among them, five gene groups associated with electron transport and 14 genes associated with root anatomical structure development were all down-regulated, and 11 genes associated with root anatomical structure development were up-regulated (Table 4).

At DS 7 d, the molecular function of Xindong23 mainly included transferase activity (59%), nicotianamine synthase activity (23%), and glycerophosphodiester phosphodiesterase activity (22%) (Fig. 5D). Among them, all DEGs associated with nicotianamine synthase activity were down-regulated, and all genes associated with phosphodiesterase activity were up-regulated under LP compared with those under CP (Table 4).

In the cell component category, 46% of DEGs were associated with respiratory chain reaction, 32% were associated with respiratory chain complex, and 22% were associated with mitochondrial membrane part (Fig. 5E). Among them, 13 genes associated with respiratory chain complexes were all down-regulated (Table 4).

Effects of two phosphorus levels on the element content of wheat roots under drought stress

The changes P and Ca content in two varieties were similar from DS 7 d to RW 3 d. The content of P and Ca were significantly higher under CP than under LP (Figs. 6A, 6B). The content of Zn was significantly lower under LP than under CP on all sampling days in Xindong23 (Fig. 6C). In addition, among the four nutrients studied, the content of K was the highest in the two varieties under LP and CP (Fig. 6D).

Figure 6 Effects of two phosphorus levels on the element content of wheat roots under drought stress.

(A and B) Histogram of each element content in Xindong20 and Xindong23 roots. (C) Histogram of Zn element content in Xindong20 and Xindong23 roots. (D) Histogram of K element content in Xindong20 and Xindong23 roots. CP: 1.0 mmol/L; LP: 0.05 mmol/L; Values are means ± standard deviation of three repetitions; one asterisk (*) and two asterisks (**) are significantly different at p < 0.05 and p < 0.01, respectively. XD20CP: Xindong20 under conventional phosphorus level; XD20LP: Xindong20 under low phosphorus level; XD23CP: Xindong23 under conventional phosphorus level; XD23LP: Xindong23 under low phosphorus level.

qRT-PCR analysis

Genes were randomly selected from two wheat varieties for qRT-PCR analysis to verify the accuracy and reliability of gene chip data. The results showed that the changes in gene expression in qRT-PCR and the chip were roughly similar, and the expression trend was very similar to the chip data (R2 = 0.73, p < 0.05) (Fig. 7), confirming that the results of the gene chip were accurate and reliable.

Figure 7 Validate gene chip results with qRT-PCR.

(A) Five random genes were selected from XD20 and XD23. Bars represent the standard errors (mean ± SE) (n = 3). (B) Relationship between qRT-PCR and chip results of randomly selected genes. Value is the log2 ratio (LP/CP) of the gene. The correlation coefficient (R2) is shown in the figure. Three biological replicates for each PCR reaction.

Discussion

Root play a vital role in wheat responses to biotic and abiotic stresses. Previous studies showed that the absorption of P and water by plants were closely related to root morphology (Schiefelbein & Benfey, 1991; Osmont, Sibout & Hardtke, 2007). In this study, under drought stress, the root growth was better under LP than under CP in Xindong20. Under LP, the total P content in the roots of the two varieties was increased at RW 3 d compared with those at DS 7 d. Meanwhile, the decreases in root vitality and root absorption area induced by drought stress were suppressed under LP. Because P could promote the metabolism of plants, the rapid increase in P content could promote the recovery of plant physiological functions re-watering.

Drought stress not only suppresses the growth of wheat root, but also the growth of shoot (Liu & Stützel, 2004). In this study, under drought stress, the root water content of the two varieties were significantly lower under LP than under CP, but the shoot water content was on the contrary. This indicates that transport of water to wheat shoot could be promoted by LP under drought stress. At present, a variety of signal sensors have been used in the studies on how plant roots sense and respond to fluctuations in water and nutrients in the environment. Ca2+ is the common second messenger (Loro et al., 2016). Previous study showed that the response of Ca2+ Arabidopsis root tips to abiotic stress could be inhibited by P starvation (Matthus et al., 2019). In this study, under drought stress, the content of Ca in the roots of Xindong20 and Xindong23 was significantly lower under LP than under CP. Based on the above, in the arid regions, relatively low P application could promote the absorption of water and P by wheat root and stimulate wheat growth. However, the genotypic differences could not be ignored. The effect of low P supply on allevating drought stress was more evident in the P-sensitive wheat variety.

The microarray results showed that there were significant genotypic differences in the response to the two phosphorus levels under drought stress. The DEGs in Xindong20 (with strong P utilization efficiency and drought tolerance) were more than those in Xindong23. It indicates that more genes are regulated to improve the drought tolerance in Xindong20. This also confirms the characteristics of the two winter wheat varieties.

Plant P transporters play an important role in the uptake and redistribution of P (Brown et al., 2012). OsPT1 is involved in the direct uptake of P from the soil, and OsPT8 plays a role in redistributing P (Nagarajan et al., 2011; Sun et al., 2012; Li et al., 2015). In this study, PT1 and PT8 were up-regulated in Xindong20, and PT8 was also up-regulated in Xindong23. This is consistent with the study results of the total P content of two varieties. In addition, K and P nutrition were closely linked (Kellermeier et al., 2014). Ródenas et al. (2017) found that in tomato and Arabidopsis, P deficiency could suppress the uptake of K and affect the expression of AKT1. Besides, the absorption of P by plants could be suppressed by high K+ content, resulting in the up-regulation of the gene associated with the root P absorption system (Ródenas et al., 2019). In this study, at DS 7 d, among the four nutrients studied, the content of K was the highest in two varieties under LP (Fig. 6D), and the content of P in the root were significantly reduced under LP. In addition, the increase of K content in Xindong23 root also confirmed the up-regulated expression of PT8 (Fig. 6D). Under LP, the total P content of root was significantly higher than that of shoot in Xindong20 at DS 7 d. The total P content of root in Xindong23 at DS 7 d was significantly compared with that at DS 5 d (Fig. 2). This may be related to the adaptability of wheat to low P supply and the regulation of P. Previous studies found that the WRKY-type bound to the W-box in the PHT1 promoters, and the expression of PHT1 was regulated by WRKY (Wang et al., 2017). Zn and P, needed by plants to maintain their essential functions (Sinclair & Krämer, 2012), could regulate the growth of plants (Zhu, Smith & Smith, 2001). The microarray results showed that WRKY19-b was up-regulated in Xindong20 and down-regulated in Xindong23. It may indicate that PT8 is down-regulated by WRKY19-b. This result is similar to the report of Devaiah, Karthikeyan & Raghothama (2007). In Arabidopsis thaliana, Zn starvation leads to the lowered expression of PHT1;1 in root (Jain et al., 2013; Khan et al., 2014). Devaiah, Karthikeyan & Raghothama (2007) found that AtPHT1;1 and AtPHT1;4 were down-regulated when the expression of the WRKY75 was down-regulated. Wang et al. (2014) and Ding et al. (2016) also obtained similar results. They found that the down-regulated expression of WRKY45 could lead to the down-regulation of AtPHT1;1. Meanwhile, Dai, Wang & Zhang (2016) showed that the overexpression of WRKY74 promoted the growth of rice rhizomes. In this study, the content of Zn in the roots of Xindong20 and Xindong23 were lower under LP than under CP at DS 7 d (Fig. 6C). Combined with the expression of WRKY19-b in two varieties (up-regulation in Xindong20 and down-regulation in Xindong23), maybe it could explain why the root growth of Xindong20 was better than that of Xindong23 under LP.

Silicon is important for improving the drought tolerance of wheat under drought stress (Gong et al., 2005). Agarie et al. (1998) showed that the membrane permeability and transpiration rate in rice were reduced by silicon application during polyethylene glycerol-induced water deficit. Gong et al. (2003) showed that silicon application could improve the drought tolerance and increase dry matter of wheat plants under drought stress compared to no silicon application. In this study, after re-watering, the root water content of Xindong20 was higher than that of Xindong23. Meanwhile, the microarray results showed that the gene associated with silicon transporterwas up-regulated in Xindong20 and down-regulated in Xindong23. Liang et al. (2003) found that silicon application could enhance the activity of superoxide dismutase (SOD) in the leaves, as well as the activity of SOD and peroxidase (POD) in the roots of barley under salt stress, thereby protecting plant tissue from oxidative damage and improving antioxidant capacity. Gong et al. (2005) also found that silicon application could reduce oxidative damage in wheat and enhance drought tolerance. In this study, in Xindong20, the expression of the gene associated with POD activity was up-regulated. This is consistent with the characteristics of the Xindong20 (high drought tolerance and P utilization efficiency).

The dry matter accumulation of wheat greatly depends on the photosynthesis. Triose phosphate is an intermediate product of photosynthesis. Macgregor et al. (2008) found that sucrose phosphate synthase was a key enzyme in the synthesis of sucrose from propyl phosphate, and its activity was affected by drought stress. In this study, the gene group associated with sucrose phosphate synthase were up-regulated both in Xindong20 and Xindong23. The accumulation of sucrose or starch in plant leaves alters the distribution of dry matter in the root, causing the changes in root structure and growth (Hermans et al., 2006). In this study, at DS 7 d, the difference in root dry weight between LP and CP treatment was not significant within the same variety. However, the difference in Xindong23 was more significant than that of Xindong20. The indicates that the effect of the external environment (water and P) on dry matter accumulation might also depend on the genotype differences.

The glycolysis process is one of the important metabolic pathways for plant. Previous study found that under a low P condition, the ATP-dependent phosphoglycerate kinase and NAD-dependent P dehydrogenase in the glycolytic metabolic pathway were affected, so that the glycolytic pathway decomposed sugar by inducing the synthesis of inorganic pyrophosphate enzyme to provide energy for growth and improve P utilization efficiency. Besides, UDP-glucose pyrophosphatase was also affected by the low P condition, which made the conversion of sucrose dependent on the pyrophosphate pathway. Pyrophosphate was catalyzed by inorganic pyrophosphatase to P while releasing a large amount of energy for use (Ciereszko et al., 2001). This study obtained similar results. In this study, the expression of the gene associated with inorganic pyrophosphatase was up-regulated in Xindong20. This may be one of the reasons why the P utilization efficiency of Xindong20 is higher than that of Xindong23.

Bernardo et al. (2017) found that under drought stress, the biosynthesis of soluble sugar, permease, and fructan in plants were increased to maintain normal growth. 6-SFT (fructan 6-fructosyltransferase) is a key enzyme involved in fructan biosynthesis and the central reserve of carbohydrates. It has an osmotic protective effect on cell membranes under drought stress conditions (Livingston, Hincha & Heyer, 2009). In this study, the expression of the gene associated with sucrose: 6-SFT was up-regulated in Xindong20, which could improve the biosynthesis of fructan. This could explain why Xindong20 has higher drought tolerance than Xindong23 to some extent.

Trehalose is involved in the responses to many abiotic stresses in plants. Trehalose-6-phosphate synthase (TPS) could affect the biosynthesis of trehalose for it plays a crucial role in the metabolism of glycolysis. In Arabidopsis, trehalose plays a critical role in the regulation of glucose and ABA signaling (Avonce et al., 2004). Iordachescu & Imai (2008) demonstrated that the plant tolerance to abiotic stresses was enhanced by the accumulation of trehalose. In this study, the down-regulation of trehalose-6-phosphate synthase in Xindong23 may lead to the repression of trehalose biosynthesis. This might be related to the poor drought tolerance of Xindong23.

In higher plants, ribonuclease plays an important role in degrading nucleic acid to release P. D’Alessio & Riordan (1997) found that the gene associated with ribonuclease degrades was up-regulated in roots under P starvation. In this study, the chip data showed that the gene expression associated with ribonuclease was up-regulated in Xindong23. This might be due to the deterioration of nucleic acid degradation in the cell of Xindong23 under drought stress. Under drought stress, the total P content in Xindong23 continuously decreased under LP. This indicates that drought stress may suppress the uptake of P by winter wheat.

Conclusions

This study revealed the mechanism of wheat responses to low and conventional P levels under drought stress by a complex regulatory network. Under drought stress, the nutrient uptake by wheat root could be improved to some extent by the adjustment of physiological indexes, ion, and gene expression under a low P condition. This could reduce the effects of drought stress on wheat growth. Besides, there was certain genotypic differences in this effect. Xindong20 showed a better root growth in response to drought stress compared with Xindong23 under LP and CP. The genes associated with silicon transporters, phosphate transporters, sucrose synthesis, glycolysis process, carbohydrates synthesis, etc., were mostly up-regulated in Xindong20, while the genes associated with the electron transport chain and the respiratory chain, glycolysis process and degrading nucleic acids were mostly down-regulated in Xindong23. Therefore, under combined drought and P stresses, maintaining the cell structural integrity and reducing the energy metabolism in wheat roots which will help to improve the drought tolerance of wheat. In future study, we will explore the effects of low phosphorus treatment on wheat root growth at different concentrations of PEG-6000 (0%, 10%, 15%, and 25%) and the key genes in wheat responses, to further explain the effects of low phosphorus treatment on the growth of wheat under different levels of drought stress.

Abbreviations

P Phosphorus

Ca Calcium

K Potassium

Zn Zinc

DEGs Differentially expressed genes

CP Conventional–phosphorus level

LP Low-phosphorus level

FDR False discovery rate

GO Gene Ontology

SOD Superoxide dismutase

POD Peroxidase

6-SFT Fructan 6-fructosyltransferase

TPS Trehalose-6-phosphate synthase

Supplemental Information

Supplemental Information 1 Completed MIAME checklist of GeneChip database.

A completed MIAME checklist for Gene expression data of winter wheat root under drought stress for 7 days by two phosphorus treatments.

Click here for additional data file.

Additional Information and Declarations

Competing Interests

Author Contributions

Microarray Data Deposition

Data Availability

The authors declare that they have no competing interests.

Xiangchi Zhang performed the experiments, analyzed the data, prepared figures and/or tables, authored or reviewed drafts of the article, and approved the final draft.

Chao Li performed the experiments, analyzed the data, prepared figures and/or tables, authored or reviewed drafts of the article, and approved the final draft.

Weidan Lu performed the experiments, analyzed the data, prepared figures and/or tables, authored or reviewed drafts of the article, and approved the final draft.

Xiaoli Wang performed the experiments, analyzed the data, prepared figures and/or tables, authored or reviewed drafts of the article, and approved the final draft.

Bin Ma performed the experiments, analyzed the data, prepared figures and/or tables, authored or reviewed drafts of the article, and approved the final draft.

Kaiyong Fu performed the experiments, analyzed the data, prepared figures and/or tables, authored or reviewed drafts of the article, and approved the final draft.

Chunyan Li conceived and designed the experiments, authored or reviewed drafts of the article, and approved the final draft.

Cheng Li conceived and designed the experiments, authored or reviewed drafts of the article, and approved the final draft.

The following information was supplied regarding the deposition of microarray data:

Data is available at the NCBI GEO database: GSE189698.

The following information was supplied regarding data availability:

The GeneChip data is available at NCBI GEO: GSE189698.

The physiological data is available at figshare: Zhang, Xiangchi; Lu, Weidan; Wang, Xiaoli; Ma, Bin; Fu, Kaiyong; Li, Chunyan; et al. (2021): Data of physiological. figshare. Dataset. https://doi.org/10.6084/m9.figshare.17085479.v2.

The morphological data is available at figshare: Zhang, Xiangchi; Lu, Weidan; Wang, Xiaoli; Ma, Bin; Fu, Kaiyong; Li, Chunyan; et al. (2021): Data of morphological characteristics. figshare. Dataset. https://doi.org/10.6084/m9.figshare.17075117.v2.

The element content data is available at figshare: Zhang, Xiangchi; Lu, Weidan; Wang, Xiaoli; Ma, Bin; Fu, Kaiyong; Li, Chunyan; et al. (2021): Element content data of winter wheat root under drought stress by two phosphorus treatments. figshare. Dataset. https://doi.org/10.6084/m9.figshare.17091116.v1.

The gene expression (qRT-PCR) data is available at figshare: Zhang, Xiangchi; Lu, Weidan; Wang, Xiaoli; Ma, Bin; Fu, Kaiyong; Li, Chunyan; et al. (2021): Gene Expression Data. figshare. Dataset. https://doi.org/10.6084/m9.figshare.17085485.v1.

The raw scanner image of GeneChip is available at figshare: Zhang, Xiangchi; Lu, Weidan; Wang, Xiaoli; Ma, Bin; Fu, Kaiyong; Li, Chunyan; et al. (2021): Raw scanner image of GeneChip. figshare. Figure. https://doi.org/10.6084/m9.figshare.17091149.v1.

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
