# Peer review of "Comparative analysis of combined phosphorus and drought stress-responses in two winter wheat"

_PeerJ, doi:10.7717/peerj.13887_

## Round 0.1 · original submission · Major Revisions

Dear Dr. Chunyan Li and co-authors,


As you can see, our reviewers found that your study was interesting and important, however they provided several comments and suggestions to strengthen your manuscript.

Reviewers 1 and 2 acknowledged the significant value of your work and had minor essential comments. Reviewer 3 pointed out some important and critical issues about the experimental design, including need to add appropriate control samples, and the validity of the findings.

I agree with almost all the comments and suggestions. I would like to ask you to address or to respond with reasons not to follow the suggestion made by these reviewers.


Best regards,
Atsushi Fukushima

·

Basic reporting

Low soil phosphorus levels and drought conditions are important constraints for wheat production in most of the areas where this crop is grown. In this manuscript the authors report the well explanation for the adaptation mechanism of wheat to drought stress under sufficient and insufficient phosphorus supply.

Experimental design

In the manuscript, the phosphorus level of sufficient and insufficient phosphorus supply was set reasonably. Two wheat materials are different. The research indicators are hierarchical and can better explain the theme of the manuscript.

Validity of the findings

In the manuscript, the experimental design is reasonable, and the specific sampling method, biological repetition, test details and data processing method can be described in detail in the data processing section. The section of transcriptome (microarray) analysis can explain the data screening method in detail, and present the results of clustering and principal component analysis. The results section is objective, and the discussion section makes a more detailed analysis and discussion on the basis of the existing literature.

Additional comments

I feel that this manuscript will be of interest to researchers involved in wheat breeding and abiotic stress in cereal crops and model plants. As a reviewer, I give my opinion on minor revision of the manuscript as a whole. Some comments which should be addressed for revision:

1-Acronyms appearing for the first time in the abstract and introduction, such as DGE and HPR, should adopt the full name.
2-“0d, 3d, 5d, 7d”, There should be a space between time and units.
3-“0d/ 3d/ 5d/7d under drought stress” etc. These words for time treatment are long and occur for many times. Consider using abbreviations.
4-Some references are incorrectly identified.
5- The latin name of Plants and the name of genes should be presented in italic form.

Reviewer 2 ·

Basic reporting

The manuscript deals with a relevant subject to PeerJ. The ms is very interesting, well written, with a relevant set of well-presented results from a rigorous investigation. Moreover, ms presents adequate discussion, the appropriate topics are supported by the literature, and the conclusions are well stated. I recommend that the manuscript should be accepted after minor revision.

Specific comments

1. Abstract: authors include the full meaning of DEGs.
2. Line 43: authors should include other aspects and remove etc.
3. Lines 47-49: references should be included.
4. Lines 50-52: authors should reformulate the sentence, as the contradiction is evident for drought-tolerant plants.
5. Authors should reformulate parts of the introduction section: for example, the beginning of sentences (lines 74 and 78).
6. Line 82: full meaning of PHR.
7. Line 121: authors should provide information about osmotic potential of PEG solution.
8. Why authors did the evaluation of water content and not of relative water content?
9. Figure 1: units of length are wrong.
10. Figure 6: element concentrations should be expressed in absolute terms (mg g-1) or, at least, the absolute values should be presented in the results section in order to have a more correct idea of mineral nutritional status.

Experimental design

No comment.

Validity of the findings

No comment.

Additional comments

No comment.

Reviewer 3 ·

Basic reporting

Language improvement necessary.

Experimental design

Experimental design is appropriate, however it requires some modifications and addition of control samples.

Validity of the findings

Do not fully agree with the conclusions, Requires improvements

Additional comments

The manuscript Comparative analysis of combined phosphorus and drought stress-response in two winter wheat varieties” brings valuable information about response of wheat under low phosphorus and drought stress. I appreciated the efforts of the author to prepare the manuscript. However, the quality of the manuscript can be improved by a major revision, here I have some suggestion.

Abstract
Line 15 – winter wheat ..Varieties .. Xindong20…………
Line 15 – “Solution cultured”. Please clarify
Line 17 – Remove word “respectively”
Line 20 – K - Postassium (full form/name for the first time and abbreviation later)
Line 21 – P , Ca , Zn ,, (full form/name for the first time and abbreviation later check throughout the text)
Line 22 – Respectively ?? Please clarify
Line 23 – The Genechip analaysis of root detected 4577 and 202 differentially expressed genes (DEGs) from Xindong20…………….
Line 27 – more effect ? Please clarify
Line 27-30 – Please specify the drought tolerant and sensitive variety. Or Rewrite the sentence Line 31 – remove Seedling stage, ( you may use abiotic stress instead)


Introduction
The introduction sum up valuable information, however, it needs to be organized in terms of the flow of information and framing of sentences overall.

Line 33 – ……….critical food crops and it development and …………
Line 35 – distribution of soil phosphorus …….
Line 46 – Mycorrhizal symbiosis is closely associated with the interaction ……
Line 48 – Protons ??
Line 55- oxygen species were accumulated in plant roots ……
Line 70-71 – Please elaborate or remove
Line 72-74 – Please clarify
Line 78-80 – What is the relevance of this information ? please clarify
Line 80 – Pi ? please use same abbreviation
Line 82 – PHR ?
Line 83 – Pht1;1?

Materials and methods

Line 111 – use the term hydroponic system instead of cultured in solution
Line 129 – Please clarify the technical and biological replicates used in the experiment
Line 133-140 – The Method used is not clear, please rewrite and provide more information
Line 139,144,152,158 – (set 3 repetitions) please use the correct method to represent the number of experimental replications. Please mention only once the number of biological and technical replicates used.
Line 151-155 – Rewrite.
Example – The water content of the wheat seedlings was determined by the method described by Gao; Briefly, the seedlings were washed under running tap water, dried using the absorbent paper, leaves and roots were separated and weighed. The samples were then dried at 105oC for 30min followed by 70oC for …min and weighed. The experiment was conducted using 3 biological replicates and 10 wheat seedlings were included in each replicate.
Line 156-161 – Rewrite and clarify with more information

Results
Line 220 – Figure 1 – Please mention the description of what are the figure A , B, C,……. In the figure legend.
Figure 1 – Please include the controls in the experiment when no drought stress was applied to samples in CP and LP
Figure 1A – What length is it? Root length? Total root length ? Please specify. Same for surface area (fig 1c)

Line 217-253 and figure 1 and 2

Please explain why the total root length reduced at day 5 and 7, Ideally the root length will not get smaller with time amongst the same samples, drought stress might only slow down the growth. Additionally, please verify if the data represented, example fig1A length of XD20CP at day 0 is ~250cm and at day 3 is ~750cm, I really doubt if wheat was able to grow at this rate.

I understand that authors are interested in drought stress, however it is important have control samples of both wheat varieties when no drought stress is applied under respective phosphorus levels. This information of control will provide valuable inputs; hence I strongly recommend to do it.

Figure legends should be clear and informative, providing correct information of what is represented in each figure.

Line 310 – 323 , Figure 6
Control experiment when no drought stress is applied is required.

Line 324-329
Although I understand the purpose of qPCR in this study was validation of RNA data. However, I would suggest to use a set of well-known drought responsive genes instead/along with random gene selected. More number of genes are required to validated the date 5 genes are not enough.
Figure 7 A – What are the values on the y axis? What do bars represent? Please specify

Line 368-370 – please elaborate.
Line 374 – Confirmed? The result only suggests the upregulation of PT1 and PT8 as an explaination for high K.

Line 398-402 – Silicon explanation is not satisfactory, please provide more information to support the results OR remove.

Line 403 – 407 – Needs more discussion.

Line 459 -468 – Needs more information and the key findings of the work based on RNA data and other experiments performed.

General comments

Authors use the word ‘And’ multiple times at the beginning of the sentence which needs to be corrected throughout the manuscript

There are several typos, grammar mistakes hence, language improvement necessary.

---

## Round 0.2 · Minor Revisions

Dear Dr. Chunyan Li and co-authors,

Thank you for your revision. However, Reviewer 3 still had essential comments for revised version. Would you please consider the experimental design to revise the manuscript?

Best regards

Reviewer 3 ·

Basic reporting

After the revision the quality of the manuscript was improved. I appreciate the authors efforts to improve the manuscript.

Experimental design

Experimental design is appropriate however, it could have been improved.

Validity of the findings

The findings look fine based on the experimental setup, however, there is room for improvement.

Additional comments

Authors explain that not including control (No drought stress) was because the experiment was designed based on the arid environment. This explanation could only be accepted if all the experiments were conducted on the field (arid environment) while in the above manuscript all the experiments were conducted under controlled environment conditions. Hence, I do not agree with this explanation.
Additional, the experiments involving abiotic stresses are always designed based on the stresses the environment brings to the plant system compared to normal conditions, the differences of the comparative analysis can be clear if compared the plants under normal environmental conditions.

---

## Round 0.3 · Minor Revisions

Dear Dr. Chunyan Li, Dr. Cheng Li, and co-authors,


Thank you for your revision. However, our Section Editor has commented and said:

"A few details needed before publication:

1) What was the genechip used? There is a standard name for each type of Affymetrix chip manufactured

2) Please list the number of replicates used for the RNA chip experiment (it is on GEO, thank you but please include it in the manuscript

3) I am confused by the GO terms "Assembly of complex" and "organization of complex". I don't think those are actual GO terms. Which complex is being referred to?"

Would you please revise the manuscript?


Best regards

·

Basic reporting

no comment

Experimental design

no comment

Validity of the findings

no comment

Additional comments

I think the paper to be overall well written and much of it to be well described. I felt confident that the authors performed careful data processing. Therefore, I recommend that a minor revision is warranted. It is suggested that the manuscript can be accepted directly.

Reviewer 2 ·

Basic reporting

No comment

Experimental design

No comment

Validity of the findings

No comment

Additional comments

I have read the new version of the manuscript and the authors thoughts on my previous remarks. I believe that the manuscript can be accepted for publication.

---

## Round 0.4 · accepted · Accept

Dear authors,

Thank you for your revision.

Best regards